# On the Security of Offloading Post-Processing for Quantum Key Distribution

**DOI:** 10.3390/e25020226

**Published:** 2023-01-24

**Authors:** Thomas Lorünser, Stephan Krenn, Christoph Pacher, Bernhard Schrenk

**Affiliations:** 1AIT Austrian Institute of Technology, Giefinggasse 4, 1210 Vienna, Austria; 2fragmentiX Storage Solutions GmbH, Plöcking 1, 3400 Klosterneuburg, Austria

**Keywords:** quantum key distribution, post-processing, secure offloading, secure outsourcing, information reconciliation, privacy amplification

## Abstract

Quantum key distribution (QKD) has been researched for almost four decades and is currently making its way to commercial applications. However, deployment of the technology at scale is challenging because of the very particular nature of QKD and its physical limitations. Among other issues, QKD is computationally intensive in the post-processing phase, and devices are therefore complex and power hungry, which leads to problems in certain application scenarios. In this work, we study the possibility to offload computationally intensive parts in the QKD post-processing stack in a secure way to untrusted hardware. We show how error correction can be securely offloaded for discrete-variable QKD to a single untrusted server and that the same method cannot be used for long-distance continuous-variable QKD. Furthermore, we analyze possibilities for multi-server protocols to be used for error correction and privacy amplification. Even in cases where it is not possible to offload to an external server, being able to delegate computation to untrusted hardware components on the device itself could improve the cost and certification effort for device manufacturers.

## 1. Introduction

Quantum key distribution (QKD) was invented almost 40 years ago and is currently a more vital field of research than ever. With commercial impact on the horizon, application of QKD is gaining substantial momentum, and the technology is expected to be deployed on a large scale in the upcoming years. This is true for both terrestrial as well as space-based applications.

QKD is the only known information-theoretic secure primitive for key exchange and can be considered part of the quantum-safe toolbox to build long-term secure information and communication technology (ICT) which even resists quantum computer threats. However, its wide adoption is still hampered by various challenges which have to be overcome to make QKD practically relevant and facilitate commercial adoption. On the one hand, research is thus continuously improving protocols, optics and electronics to achieve a better bandwidth and distance, as well as co-existence with existing infrastructure. On the other hand, miniaturization and electro-optical integration are important topics to make the technology more reliable and cost-effective.

Complementary to these efforts, our work focuses on the possibility to offload (outsource) computationally expensive tasks in the QKD post-processing phase to the external infrastructure without compromising the overall security. Being able to outsource these tasks to external data centers allows for simpler and less power-hungry devices in the field, resulting in more versatile applications for QKD.

### 1.1. Related Work

Improving the efficiency and throughput of the post-processing phase is still an interesting challenge in the context of QKD. Scientific and industrial research initiatives are focusing on algorithmic improvements to reduce computational effort (c.f. [1,2,3,4]) or extending the local computational resources with specialized hardware. They leverage equipment from high-performance computing or graphics processing units [5,6,7] or even develop dedicated hardware designs as co-processing units for local installation [8,9].

### 1.2. Contributions

In this work, we contribute to these efforts via a complementary approach by presenting novel methods for offloading (or outsourcing) the most expensive parts of the QKD post-processing stack. For this, we combine our expertise from QKD and cryptography. We will clearly demonstrate the problem, show the benefits of our approach and present new protocols and barriers.

More precisely, we present and analyze protocols for outsourcing information reconciliation to external and untrusted environments, therefore facilitating new application scenarios (e.g., usage in low-power access networks). We furthermore discuss possibilities to outsource the privacy amplification step, which could further help to reduce the required processing power in QKD nodes. Additionally, we present possible use cases to undermine the practical relevance of the novel developed methods.

### 1.3. Outline of the Work

In Section 2, we present and discuss quantum key distribution and the required steps for post-processing, as well as the motivation for offloading computationally expensive tasks. In Section 3, we review information reconciliation in detail and present a new protocol which allows its outsourcing in a secure way as well as an impossibility result. In Section 4, we analyze the potential of outsourcing privacy amplification and propose new methods in this direction. Potential use cases for the proposed solution are then discussed in Section 5, and the concluding remarks are given in Section 6.

## 2. Quantum Key Distribution

Contrary to most other cryptographic primitives, QKD is a cryptographic key-agreement protocol which derives its security from the physical layer (i.e., it uses a quantum channel to exchange quantum information which cannot be perfectly copied or eavesdropped according to the laws of quantum mechanics). In preparing and measuring the QKD protocols, so-called quantum bits (qubits) are encoded and transmitted over a quantum channel. Typically, the qubits are encoded on photons, and the transmission channels are either fiber optics or free space. Finally, the qubits are measured at the receiver and decoded. From the measurement of quantum bits, classical information is derived, and all of the following steps are carried out in the classical domain. However, due to their interaction with the environment and eavesdroppers, photons are subject to perturbation and absorption. To detect and cope with these modifications in the transmission channel, post-processing steps have to be applied in order to obtain the full key agreement primitive with practical correctness and secrecy guarantees.

The outstanding property of QKD is that it is an information-theoretic secure (ITS) and universally composable (UC) key agreement protocol [10], given that its classical communication is performed over an authentic channel (note that all key-agreement protocols are insecure over non-authentic channels). ITS message authentication codes are based on universal hashing [11], which in the first round uses pre-shared keys, and in later rounds, QKD keys from previous rounds are a means to generate an ITS authentic channel [12]. Thus, QKD is a very powerful cryptographic primitive which cannot be realized with non-quantum protocols.

### 2.1. QKD Post-Processing

QKD comprises two phases to arrive at a key agreement with strong correctness and secrecy guarantees. First, qubits are randomly generated on one side, transmitted over the optical quantum channel, and measured on the other side to generate the so-called raw key. In the second phase, a non-quantum (classical) post-processing protocol is executed to agree on identical keys (correctness) on both ends of the transmission line and to render useless any information a potential attacker could have learned by attacking the transmission phase (secrecy).

In detail, the steps to extracting a secure key from the raw data of the transmitted quantum bits are as follows:**Sifting** removes non-relevant information from the raw key (e.g., in conjugate coding protocols, events prepared and measured in different bases are deleted). Additionally, events not received by Bob are discarded in discrete-variable protocols (cf. Section 3).**Error estimation** determines an upper bound on the information leaked to an adversary on the quantum channel and can provide information to optimize the subsequent information reconciliation. Although more advanced methods have been proposed in the literature, this is typically accomplished by cut-and-choose methods. Additionally, the idea of using a confirmation phase to replace error estimation was proposed by Lütkenhaus [13].**Information reconciliation**, which often uses methods from forward error correction, aims at correcting all errors in the remaining raw key so that the sender and receiver should obtain identical keys. The classical (non-quantum) messages exchanged in this process must not leak information on the final key. Typically, the leakage is tracked and treated during the privacy amplification step.**Confirmation** detects non-identical keys (for which information reconciliation has failed) with a probability close to one. If non-identical keys are detected, then the parties either go back to the information reconciliation step or abort the QKD protocol.Finally, **privacy amplification** eliminates the information leaked during all protocol steps (quantum and classical) from the final key by running a (strong) randomness extraction protocol between the peers.

All processing steps together enable Alice and Bob to agree on a final key which is ϵ-close to an ideal key. Various optimizations of the above key agreement process have been proposed in the past, either for efficiency reasons or implementation aspects, but this generic structure is typically followed in one way or another.

### 2.2. Motivation to Offload Post-Processing

From a computational perspective, information reconciliation is by far the most expensive task computationally in the stack and can limit the throughput in high-speed systems [14,15]. The second-most computationally demanding task is privacy amplification [6]. The rest of the protocol steps are rather simple tasks and can be executed in real time even on embedded platforms.

Therefore, we introduce and study the idea of offloading these tasks from devices by outsourcing computation to untrusted or less trusted hardware in an ITS way. The ability to outsource information reconciliation (IR) and potentially privacy amplification (PA) would enable new application scenarios for both access networks and transmission systems. On top of this, satellite-based QKD could become more versatile if processing resources can be shifted around more easily.

The two main advantages gained by offloading processing to external hardware are increased efficiency and flexibility in the use of compute resources, also resulting in a higher energy efficiency, and a reduced attack surface by limiting the number of components dealing with secure key material.

QKD systems are deployed for long-term security and produce large capital expenditure (CAPEX) spending (i.e., they are used over a long period of time). Putting all the processing power into the devices at build time hinders later updates and prevents the operator benefiting from Moore’s law. If the hardware is outsourced, then it could be updated during the lifetime of the system with new technologies, resulting in further optimized efficiency. Furthermore, if the hardware need not be trustworthy and certified, then cheaper commercial off-the-shelf (COTS) hardware can be used. It would even be possible to completely outsource this to public cloud infrastructures in an extreme case.

Additionally, time sharing allows for further improvements in certain use cases. If QKD is used in hybrid encryption protocols to establish session keys [16], then high key rates are not needed, and sharing the computational resources between links can further reduce the CAPEX and also the operational cost (OPEX). Hence, putting the computationally expensive tasks into efficient data centers which do not even need to be trusted is very desirable. This allows for hardware updates and joint management of all QKD workloads in the field with its continuous upgrading probabilities, which is especially favorable for operators of QKD networks. Because the communication overhead is minimal compared with the computational one, a clear advantage in terms of energy efficiency arises, and the gained flexibility in managing tasks is very advantageous.

Furthermore, the proposed approach could also be used within a system. Treating parts of the system as untrusted could eventually provide the possibility for system updates without compromising system certification and help to reduce the OPEX cost during the system’s lifetime.

## 3. Outsourcing Information Reconciliation

As mentioned in Section 2, information reconciliation (IR) is the most demanding task in post-processing of QKD, independent of the protocols being used on the quantum level. Error correction is computationally intense because of high error rates encountered in combination with the constraints on the amount of information disclosed during error correction. The information revealed during the public discussion must be kept as short as possible to maximize the overall system performance. Ideally, IR works close to the Shannon limit. If keys have to be processed in real time, error correction is the bottleneck of post-processing and can introduce substantial problems in resource constraint environments.

On a quantum level, QKD protocols can be divided into two classes—*discrete variable* (DV) and *continuous variable* (CV) QKD—which also result in different requirements for information reconciliation. In DV-QKD protocols (e.g., BB84 [17]), qubits are measured by single photon detectors. Due to channel attenuation and non-perfect detectors, the rate of detected photons is typically orders of magnitudes lower than the rate of prepared photons. Consequently, in DV-QKD, IR schemes must typically provide the possibility to operate on the final key rates in the order of kilobits per second [18] up to megabits per second [14]. In CV-QKD systems, signals are only perturbed and not lost through channel effects, resulting in very high raw key rates but also high error rates compared with a discrete variable system.

Additionally, two basic types of IR protocols can be distinguished in QKD systems. On the one hand, interactive two-way protocols have been developed for highly efficient correction capabilities near the Shannon limit, with CASCADE [1,19,20] being the most prominent representative. They can achieve smaller leakage than any one-way protocol, but their practical performance is limited due to their interactive nature by the latency of the classical channel. On the other hand, forward error correcting schemes have been adopted and developed further to be used in operational regimes encountered in QKD [21]. One-way IR based on low-density parity check codes (LDPC) is currently the most efficient representative in this category and is used in many prototype systems [22].

One-way schemes have many desirable properties when it comes to realization and can easily be parallelized to increase performance. Therefore, in our work and for the design of our protocols, we assume the use of forward error correction where necessary and, in particular, LDPC-based error correction in case concrete implementations are considered.

### 3.1. Linear One-Way Information Reconciliation

Before presenting our scheme for offloading, we first explain one-way IR in the context of DV-QKD in more detail and informally define the concept of secure outsourcing for IR. Traditional error-correcting block codes consist of sets of codewords that contain redundant information. Before sending data over a noisy channel, the data are encoded into codewords. The contained redundancy can then be used by the receiver to correct the introduced errors.

One-way reconciliation (i.e., source coding with side information) has been studied since the 1970s [23,24]. While related to error-correcting codes, the idea here is that the data are transmitted over a noisy channel without adding any redundancy. Rather, the source additionally sends a syndrome of the data (computed with a parity check matrix of an error-correcting code) over a *noise-free* channel. The receiver then uses the syndrome together with the noisy data (side information) to decode the original data.

More concretely, after sifting, Bob has obtained a noisy version of Alice’s sifted key (i.e., $k_{B}=k_{A}+e$, where e denotes the error vector). Alice and Bob then use a linear block code with parity check matrix H. Alice computes the syndrome of her sifted key $k_{A}$ with the help of H by computing
$$s_{A}:=k_{A}H^{⊤}.$$

Alice sends $s_{A}$ over the noise-free classical channel to Bob. Bob corrects his sifted key by (approximately) solving the problem of finding a vector ${\hat{k}}_{A}$ which, among all vectors with syndrome $s_{A}$, has the smallest Hamming distance to $k_{B}$.

Searching for this vector is computationally hard and is equivalent to solving the standard syndrome-decoding problem (NP-complete) which, given H and a vector s, requires finding a vector e of a minimal weight satisfying $eH^{⊤}=s$. The equivalence can easily be seen considering that in our case, the syndrome decoder is employed for $eH^{⊤}=k_{B}H^{⊤}-k_{A}H^{⊤}=k_{B}H^{⊤}-s_{A}$.

### 3.2. Protocol for Offloading Direct Reconciliation

In the context of QKD, if Alice, who sends the qubits, also sends the syndrome to Bob, and Bob corrects erroneous bits to obtain the sifted key of Alice as an agreed key, then the protocol is called *direct reconciliation* (DR). In DV-QKD, which is considered symmetric [25], the roles of Alice and Bob can also be interchanged during IR, resulting in so-called *reverse reconciliation* (RR). However, the same is not true for CV-QKD, where the direction of the IR protocol does matter for higher transmission rates, as will be discussed later.

We present a simple scheme for remote (outsourced) information reconciliation called REM-IR (remote IR), which allows the computationally intense step of syndrome decoding to be outsourced to an untrusted party in a secure way. The idea is to give the error syndrome (i.e., $s_{e}=s_{B}-s_{A}=k_{B}H^{⊤}-k_{A}H^{⊤}$) to an external party, which returns the error vector e with a minimal weight satisfying $s_{e}=eH^{⊤}$.

We want to emphasize that searching for $eH^{⊤}=s_{e}$ can be carried out with the very same algorithms and techniques typically used locally by Bob (e.g., efficient variants as in [21] or [26]). The problem is that finding e with a minimum Hamming weight is fully equivalent to the problem of finding a ${\hat{k}}_{B}$ with a minimum Hamming distance to $k_{B}$ which fulfills ${\hat{k}}_{B}H=s_{B}$, which is due to the linearity of the code. As an example, if LDPC codes are used which are characterized by a sparse H, then all types of decoders can be used in the very same way to find e close to the zero code word as in finding ${\hat{k}}_{B}$ close to $k_{B}$. The only difference in the two ways error decoding is applied is that in the latter, ${\hat{k}}_{B}$ is directly computed, and in the first case, it is computed by ${\hat{k}}_{B}=k_{B}+e$.

Informally, the protocol is secure because the information leaked by publishing $s_{e}$ and e in addition to $s_{A}$, and thus also $s_{B}=s_{A}-e$, does not increase the information of Eve regarding the agreed key string $k_{A}$. The intuition behind this is that just learning a bit flip vector of a largely unknown key does not increase the information about the key. Put differently, the information leaked about the final key $k_{A}$ by additionally learning e is zero because e is independent of $k_{A}$ and removed from $k_{B}$ in the IR step anyway. The described protocol is also equivalent to interactive error decoding, as introduced in CASCADE [19], which also leaks parity information and error bit locations during the public discussion.

A detailed description of the protocol is shown in Figure 1, and the security of the protocols is proven in the following:
**Theorem 1** (Security of **REM-IR**). *REM-IR is a secure scheme for offloading direct reconciliation for DV-QKD and does not leak any additional information about the agreed key by public discussion compared with a local IR (i.e., the mutual information between Eve’s key and the agreed key is the same as with local IR).*
**Proof.** Let $K_{A},K_{B}$ be *n* bit random variables representing correlated sifted keys for Alice and Bob, which are used as input to information reconciliation. $S_{A}$ is a random variable representing the syndrome computed by Alice, and E is a random variable for the error introduced on *n* channel usages. The quantum channel between Alice and Bob is then modeled as a binary symmetric channel BSC(e) with a quantum bit error probability *e*. Furthermore, let $L_{E}^{IR}\left(K|Q\right)$ be the additional information leaked to Eve about the agreed key K during the information reconciliation phase beyond what Eve already gained during the previous steps of the key exchange.Moreover, $H\left(K_{A}\right)=H\left(K_{B}\right)=n$ is for uniformly random input encoding, and mutual information $I_{AB}I\left(K_{A};K_{B}\right)=n\left(1-H_{b}\left(e\right)\right)$ is defined by the error probability on the channel. Thus, the amount of information required to be exchanged during public discussion is $\left|Q\right|\geq H\left(K_{A}|K_{B}\right)=nH_{b}\left(e\right)$, where we assume an ideal reconciliation algorithm which works at the Shannon limit (i.e, equality holds).Without loss of generality, we assume that Bob will correct his errors and the agreed key will be $k=k_{A}$. Note here that in the DV-QKD-type protocols, we have $I_{AE}=I_{BE}$ [25], which makes them suitable for direct reconciliation.Now, we show that the information Eve gains about the final key during a protocol run of REM-IR is equal to the information leaked in local IR, where it only sees $S_{A}$. Concretely, by revealing $S_{e}$ and therefore E and $S_{B}$ in addition to $S_{A}$, the information Eve learns about the final key (k) can be described as follows:
$$L_{E}^{\mathrm{REM}-\mathrm{IR}}\left(K|S_{A},S_{e}\right)=L_{E}^{\mathrm{REM}-\mathrm{IR}}\left(K|KH^{⊤},EH^{⊤}\right)=L_{E}^{\mathrm{REM}-\mathrm{IR}}\left(K|KH^{⊤}\right)=L_{E}^{\mathrm{REM}-\mathrm{IR}}\left(K|S_{A}\right)\;.$$This is due to the fact that the additional information Eve gains by learning $S_{e},E$ and therefore $S_{B}$ is only about $EH^{⊤}$, which is not correlated to the final key and also removed from $k_{B}$ during the IR step. This can also be seen by looking at Bob’s key $K_{B}=K_{A}+E$, which is the sum of two independent random variables where the error term is removed by IR and does not contribute in any form to the final key string K. Therefore, the leakage during the protocol run is $L_{E}^{\mathrm{REM}-\mathrm{IR}}\left(K|S_{A}\right)=\left|S_{A}\right|=nH_{b}\left(e\right)$. □

Variants of REM-IR could be, for example, to let Alice and Bob directly send $s_{A}$ and $s_{B}$, respectively, to the third party, who then computes $s_{e}=s_{B}-s_{A}$. This version is equivalent, as also in REM-IR, the third party knows all syndromes (i.e., it can compute $s_{B}=s_{e}+s_{A}$ from the publicly known $s_{e}$, $s_{A}$). Furthermore, to increase the reliability and availability of the results, the computation can be delegated and distributed to an arbitrary number of third parties. The security is not jeopardized by any extended protocol involving more external untrusted parties and serves as a general baseline for such scenarios.

### 3.3. On Outsourcing Reverse Reconciliation

For continuous-variable QKD (CV-QKD), we have different requirements than for discrete-variable QKD which not only impact the modulation schemes but also the information reconciliation. On the Qbit level, CV-QKD uses homodyne detection, which allows for soft or hard decoding. For simplicity, we will look only at discrete modulated CV-QKD, particularly binary modulation. Therefore, in the following, we treat the CV-QKD system as a hard-input–hard-output channel which operates on classical bit strings.

The idea of reverse reconciliation was introduced by Maurer [27] for classical communication and later applied to CV-QKD to overcome the 3 dB loss limit [28]. In essence, reverse reconciliation is based on one-way error correction in a reverse configuration, with Bob sending the syndrome $s_{B}$ to Alice and Alice correcting her bits.

The underlying model is based on two channels: one connecting Alice and Bob and the other connecting Alice and Eve. Interestingly, if reverse reconciliation is applied in this scenario, a key can still be distilled even if the channel from Alice to Eve is superior to the one from Alice to Bob. The secret capacity of the channel for reverse reconciliation in [27] was derived as $C_{s}=H_{b}\left(e+d-2ed\right)-H_{b}\left(e\right)$ when Alice and Bob had access to a broadcast channel for public discussion. The bit error probabilities are *e* and *d* for the channels from Alice to Bob and Alice to Eve, respectively, and e+d−2ed for the conceptual channel from Bob to Eve. $H_{b}$ is the binary entropy function.

In the classical model of [27], Shannon entropy is used in the analysis. For the case of CV-QKD, the mutual information between Bob and Eve has to be replaced by the Holevo information, and finite key effects have to be considered [29]. However, both refinements do not affect our treatment based on generic BSC channels.

For the secret channel capacity argument to be valid, Alice’s key has to be kept private, thus preventing Eve from correcting the error bits in her key. With this additional requirement, outsourcing information reconciliation directly, as performed in REM-IR, is not possible. If both $s_{e}$ and $s_{B}$ are leaked, then $s_{A}=s_{e}+s_{B}$ can easily be computed, and the advantage over the conceptual channel is lost because Eve can correct all errors in the string with Alice and remove the uncertainty $H(K_{E}|K_{B})$.

More formally, the following result shows that *fully offloading* error corrections (i.e., letting a third party perform the entire error correction and simply return e) to an untrusted party cannot be achieved for both classical reverse reconciliation and in the quantum setting:

**Theorem 2** (Impossibility of external syndrome decoding for classical RR). *For reverse reconciliation in the classical (non-quantum) setting, full offloading syndrome decoding is not possible with a positive key rate.*

**Proof.** Alice is connected to Bob and Eve over binary symmetric channels (BSCs) with error rates *e* and *d*, respectively. She sends out the very same signal $k_{A}$, which is received as $k_{B}$ and $k_{E}$. In the case of RR, we further have that Alice sends the signal $k_{A}$ but corrects her key for the error received by Bob (i.e., $k_{B}$ is the final key k).For binary random input encoding, it holds that $H\left(K_{A}\right)=H\left(K_{B}\right)=1$, and the mutual information $I_{AB}I\left(K_{A};K_{B}\right)=1-H_{b}\left(e\right)$ is defined by the error probability on the channel. $K_{A}$, $K_{B}$ and $K_{E}$ are the binary correlated random variables at Alice, Bob and Eve, respectively. The amount of information required to be exchanged during public discussion for reverse reconciliation per channel use is $\left|Q\right|\geq H\left(K_{B}|K_{A}\right)=H_{b}\left(e\right)$. For the proof, we assume that optimal codes reaching the Shannon limit are used (i.e., equality holds for the syndromes communicated).Thus, for offloading, any external party taking over the syndrome decoding for *n* bit keys based on a public H needs $|s_{e}|=nH_{b}\left(e\right)$ amount of information to correct for the errors on the AB channel. Note here that $s_{e}$ itself does not carry any information about the key yet still fully defines the error e.We now prove the impossibility in two steps. (1) We calculate the change in mutual information by offloading the computation of e by Alice and therefore publishing the error syndrome $s_{e}$. We then discuss the influence of discussion needed between Alice and Bob to compute the error syndrome $s_{e}=s_{A}-s_{B}=k_{A}H^{⊤}-k_{B}H^{⊤}$ in the first place, which clearly needs contributions from both peers.Furthermore, we know that Eve is not allowed to learn enough information about k to correct all errors through its conceptual channel (i.e., $I_{AB}$–$I_{EB}$ have to be preserved or at least be larger than zero to leave Alice and Bob with a secure key).In the beginning of the protocol, we have $I_{AB}=1-H_{b}\left(e\right)$ and $I_{EB}=1-H_{b}\left(e+d-2ed\right)$. After publishing $s_{e}$ and computing e in step (1), the mutual information per bit changes to $I_{AB}^{\left(i\right)}=1$ and $I_{EB}^{\left(i\right)}=1-H_{b}\left(e+d-2ed\right)+H_{b}\left(e\right)=I_{EA}^{\left(i\right)}$, respectively. This means that with knowledge of $s_{e}$ and implicitly e, Alice can correct for all errors with Bob, but Eve is left with some remaining uncertainty.Now, to compute $s_{e}=s_{A}-s_{B}$, another $nH_{b}\left(e\right)$ bits have to be communicated in advance between Alice and Bob (2), which further impacts the knowledge of Eve about the keys. However, after exchanging another $nH_{b}\left(e\right)$ bits about $k_{B}$ in public to compute the error syndrome, we still have $I_{AB}^{\left(ii\right)}=1$, but $I_{EA}^{\left(ii\right)}$ is also increased to 1 because $I_{EB}^{\left(i\right)}+H_{b}\left(e\right)=1-H_{b}\left(e+d-2ed\right)+2H_{b}\left(e\right)>1$. Alice already corrected all errors in step (1), and thus the additional information does not further increase their knowledge. On the contrary, for Eve, the information in step (2) is useful and further increases the mutual information with Bob up to the maximum of one, which means Eve has full knowledge about the agreed key. This is due to the fact that the published information is about the independent random variables $K_{E}$ and $K_{B}$ both contributing to the key agreement individually, where $k=k_{B}=k_{A}+e$. In summary, Eve either learns the key or, if step (2) is encrypted, leads to a negative key balance for QKD in the region of interest with d≤e. □

**Corollary 1** (Impossibility for quantum RR). *For reverse reconciliation in the quantum setting, full offloading syndrome decoding is not possible with a positive key rate.*

**Proof.** The quantum case is based on the same assumptions as the classical case and derives by looking at the entropies. Bob and Eve are connected to Alice over a quantum channel, where $I\left(K_{A};K_{E}\right)\geq I\left(K_{A};K_{B}\right)$ holds. We also assume a symmetric system with $H\left(K_{A}\right)=H\left(K_{B}\right)=1$, and consequently $H\left(K_{A}|K_{B}\right)=H\left(K_{B}|K_{A}\right)$ as well as $H\left(K_{A}|K_{E}\right)=H\left(K_{E}|K_{A}\right)$, due to Bayes’ theorem. We also know from the definition of mutual information that Eve has less uncertainty about the final key (i.e., Bob’s key) than Alice $H\left(K_{A}|K_{E}\right)\leq H\left(K_{A}|K_{B}\right)$. Additionally, because the entropy function is concave, we also know that $H\left(K_{A}|K_{E}\right)\leq H\left(K_{B}|K_{E}\right)\leq H\left(K_{A}|K_{E}\right)+H\left(K_{A}|K_{B}\right)$. Due to Slepian-Wolf’s theorem [23], we require Bob to communicate $H(K_{B}|K_{A})$ bits (e.g., $s_{B}$) to enable Alice to compute the error syndrome. Furthermore, we require Alice to eventually publish $H(K_{A}|K_{B})$ bits (e.g., $s_{e}$) in order to fully outsource error correction, also assuming an optimal code. Contrary to forward reconciliation, both strings published are useful for Eve because the information about the error is independent from the bits revealed about Bob’s key.Thus, with access to this public information, Eve is now able to reduce its uncertainty $H(K_{B}|K_{E})$ about the key because
$$H\left(K_{B}|K_{E}\right)\leq H\left(K_{A}|K_{E}\right)+H\left(K_{A}|K_{B}\right)<2H\left(K_{A}|K_{B}\right)\;,$$
leading to $I(K_{A};K_{E})=1$. In essence, after seeing $s_{e}$ and $s_{B}$, Eve can calculate $s_{A}=s_{B}-s_{e}$ and remove all uncertainty $H\left(K_{E}|K_{A}\right)<H\left(K_{A}|K_{B}\right)$ about $k_{A}$ and subsequently the final key $k=k_{B}$. □

However, even with these results in mind, it is unclear if weaker notions of offloading would enable certain levels of partial or assisted secure outsourcing with positive key rates. An impossibility result for *partial offloading* is hard to formalize, as in an edge case, no meaningful computation would be delegated to the untrusted server, and the entire error reconciliation would be performed as local operations. In the following, we argue that no obvious or natural approaches for reasonable (partial) delegation of computations exist.

We have seen that to be left with a secure key after all steps, the outsourced error reconciliation has to hide either $s_{e}$ or $s_{B}$ with ITS properties. However, $s_{e}$ cannot be encrypted by masking because the nature of the outsourced computation is to find a minimum weight vector which fulfills $eH^{⊤}=s_{e}$ for a given $s_{e}$ and a public H, which always requires publishing a target vector e, which is the reference for distance minimization. Therefore, a simple solution is to encrypt $s_{B}$ during transmission with previously acquired secure key material. This requires $nH_{b}\left(e\right)$ additional key bits, leading to a reduced capacity of $C_{s-enc}=H_{b}\left(e+d-2ed\right)-2H_{b}\left(e\right)$. Although the protocol is secure and enables offloading of error correction, it does not lead to a positive key rate for the regions of interest where d<e, which is also shown in Figure 2.

To give more evidence that an encrypted IR protocol with a positive key balance is not achievable, we review the most relevant and evident techniques to protect the key of Alice or even $s_{A}$ in an ITS sense to prevent Eve from learning Alice’ key or increase $I(K_{A};K_{E})$. In order to build an encrypted RR protocol, different techniques could be used, but the parity check matrix H is considered to be publicly known, which limits the application of hiding techniques to the raw key vector. Furthermore, the discussed solutions should not increase the computational effort to correct errors.

We start from the syndrome-decoding equation $s_{e}=s_{A}-s_{B}=eH^{⊤}$ and discuss options to hide $s_{A}$ from Eve or to prevent any increase in $I_{AE}$ by public discussion. In order to hide the bit flip positions, we discuss the following additional techniques, which are evident approaches toward the security goals for offloading RR but also not providing any positive key rate due to encrypting the raw key by the following means:A one-time pad (OTP);Permutation;Padding (i.e., adding dummy (error) bits).

*Encryption.* If the goal is to hide $s_{A}$ in an ITS way given that H is public, either $s_{e}$ or $s_{B}$ must be one OTP encrypted. $s_{B}$ can be encrypted when transmitted to Bob or already at the key level, therefore ultimately hiding the key $s_{B}^{\prime }=\left(k+m\right)H^{⊤}$, where m is a random masking value, which must also be securely transmitted from Bob to Alice. However, in the first case, $|s_{B}|=nH_{b}\left(e\right)$ bits are optimally required, and in the second case, a number of raw key bits |k| is required, which is extremely inefficient. Above, we have already shown that even the first case leads to negative key rates.

Unfortunately, encryption of $s_{e}$ also cannot be used to hide bit error positions because decoding requires a start vector to explore the vicinity too. Finding a vector close to a random vector with public H leaks the bit flip positions e and therefore also $s_{e}$.

*Permutation.* An alternative method to hide e, $s_{e}$ and thus $s_{A}$ would be by permuting the raw key bits before running RR with an unencrypted $s_{B}$. Using a permuted key $k^{\prime }=\Pi \left(k\right)$ for the post-processing would render error correction information useless for Eve, but it has to be random for each block and applied on both peers in secret. Thus, a huge amount of shared key material is required, given that the permutation has to be selected randomly from the n! possible ones, which requires O(nlog(n)) bits to represent. In the end, if the selected permutation has to be communicated over the public channel via OTP, the key balance is even worse than with syndrome encryption.

*Padding.* Padding the raw key with dummy bits could be used to hide error bits if combined with permutation. This corresponds to the technique of mixing raw key with dummy key bits. However, in this case as well, the positions and value of the dummy key bits have to be agreed upon secretly by Alice and Bob, which also requires too many bits.

Finally, additional errors could be introduced only to Alice. Because the remaining error margin in practical CV-QKD is already very small, this technique can only hide a small amount of information and substantially increases the computational work at the remote instance through the increased error rate.

In summary, all natural approaches for partially offloading RR with a positive key rate to a single server in general seem unfeasible.

### 3.4. Verifiability of Outsourced IR

Aside from the challenge of efficient yet secure outsourcing of information reconciliation, it is also important to have a means to efficiently check the correctness of the solution. This prevents from actively malicious behavior of the remote instance performing the actual work.

Fortunately, the problem of error decoding comes with an efficient algorithm to check the result:Check if $eH^{⊤}=s_{e}$; otherwise, abort the process.*Optional*: Check if the weight of e is indeed below the threshold of the code or is consistent with the estimated error, and abort otherwise.

The first check can be easily computed by conducting the vector matrix multiplication and only requires additions to modulo 2 (XOR) in the order of bits set in H, which is very efficient for LDPC codes. The second check is even faster if it can be performed for the used code. The Hamming weight of e must be smaller than what can be corrected by the code. However, the correction capabilities of a code cannot always be bound, especially for the often-used LDPC, where this is not possible. In such cases, only the estimated error rate can be used to test the hypothesis of a bit flip vector being correct. Nevertheless, there is still the final confirmation phase where an ultimate check is completed to assure the key error probability. However, directly verifying the IR outsourcing step enables attribution of errors to external servers and flexible reaction aside from aborting the whole process.

In summary, verifiability immediately follows from the nature of the problem. This makes protection against malicious remote servers possible with minimal effort and does not require a full recomputation by Alice.

### 3.5. Multiparty Computation-Based Outsourcing

In the previous subsections, we presented an efficient solution for offloading direct reconciliation (DR) to a single server and discussed the problems with RR. Although relying on a single untrusted server seems to be the most desirable use case, it is natural to ask how efficient a multi-server configuration would be in cases where single-server offloading is not possible. If multiple servers are available, then ITS multiparty computation protocols (MPCs) based on secret sharing —as introduced by Ben-Or et al. [30] and Chaum et al. [31]—can be used to obliviously compute arbitrary functions on sensitive data, and thus they can also be used in CV-QKD because the inputs are kept private from the servers. The respective class of MPC protocols with ITS security operates in the honest majority setting (i.e., under the assumption that an adversary corrupts less than half of the MPC-computing nodes). Aside from the non-collusion assumption, the protocols also rely on secure channels, which can be assured by different means, as discussed in Section 5.

More concretely, in MPCs, a set of parties can jointly evaluate a function without leaking any information to any of the participating parties beyond what can be derived from their own inputs and the computation result itself. Thus, an MPC provides *input secrecy* (or *input privacy*) (i.e., no party learns the input values of any other party) and *correctness* (i.e., the receiver of the result is ensured that the result is correct). In an honest majority setting with less than half of the servers being corrupt, ITS MPCs are among the most performant approaches for computing with encrypted data and achieve practical performance in many application scenarios.

We therefore looked into the problem of MPC-based information reconciliation with ITS security on the basis of secret sharing [32]. If IR is performed in MPCs, the decoding can be accomplished without learning anything about the error syndrome $s_{e}$ (private input) or error vector e (private output), but the parity check matrix can still be kept clear. In this model, the peer (Alice for RR) offloading IR is encoding the error syndrome as private input for the MPC system. The MPC system then obliviously computes the bit flip vector by executing a distributed protocol realizing a privacy-preserving LDPC decoder. The result of the computation is secretly shared among the MPC nodes after the protocol run and sent back to the peer (Alice), who can reconstruct it.

To demonstrate the feasibility of the solution, we study the practical efficiency of error decoding for low-density parity check (LDPC) codes in an existing MPC framework and estimate the performance that can be achieved. To the best of our knowledge, this is the first time this problem is considered. The only known related work was presented by Raeini and Nojoumian [33], who only considered Berlekamp–Welch decoding for Reed–Solomon codes.

In general, we distinguish two main types of message-passing algorithms for LDPC decoding: bit-flipping algorithms and belief propagation [34]. The decoding approach typically used in QKD is from the category of belief propagation (BP) and specifically uses sum-product mechanisms to update beliefs, an approach which works very efficiently on plaintext data. Unfortunately, this approach is not well suited for direct application in MPCs. This is because the algorithm works on floating point numbers and uses trigonometric functions in the belief update part, with both being very inefficient in MPCs. Thus, to initiate the research topic, we focused on bit-flipping (BF) algorithms first in our implementation approach because they seemed to be more promising, although they suffer from inferior performance in terms of information rate. BF algorithms have a very simple structure and work extremely fast (e.g., if implemented in hardware).

The bit-flipping algorithm is a non-probabilistic hard-input hard-output decoding algorithm and works on the Tanner graph representation of the code. The messages passing back and forth are all binary. The main structure of the BF algorithm is similar for all variants. In the first step, the variable nodes send their current values to the check nodes. Then, the check nodes compute their values and feed back the results to the adjacent variable nodes, signaling if the check is valid. After each variable node receives the check bits from all connected check nodes, the current guess for the code word is updated. Different approaches exist to update the variable nodes, and to the best of our knowledge, no optimized codes or methods for the particular case of QKD have been studied or analyzed. Therefore, we selected one of the most prominent solutions—Gallager’s algorithm [35]—to demonstrate feasibility and applied it to (non-optimized) codes available for BP algorithms.

We developed an MPC version for the bit flip-decoding algorithm which is shown in Figure 1, where [[·]] denotes variables which are processed in the encrypted domain and are therefore kept confidential. The implementation assures that the code word itself as well as related information is only kept in a secret shared form among the parties and never revealed during computation. On a high level, the algorithm performs message passing in a Tanner graph defined by the parity check matrix *H*, with binary variables represented by public integers with values zero and one. We encoded the bits on integers because ITS-MPCs work on algebraic circuits, with additions being compared almost for free and multiplications requiring interactions between nodes. This allowed us to quickly count the number of ones in a vector of bits and also enabled exclusivity over vectors by reducing modulo two after summing them up. The algorithm comprises four major steps which are iteratively repeated until a valid code word is found or the maximum number of iterations is reached:In the first step, the variable nodes pass their values to the check nodes, where they are combined to compute the check value, which is zero when the check is fulfilled and one if not.In the second step, the values of the check nodes are passed back to the variable nodes, where they are aggregated (i.e., the number of check nodes not satisfied is counted for each variable node).Thirdly, the algorithm terminates if all checknodes are zero.Finally, the algorithm computes which variable nodes have to be flipped. Here, we used Gallager’s algorithm B [35] in our implementation, which basically compares the counts computed in step two against a threshold value to decide which bits are flipped. Although the threshold value for comparison is public, this comparison has to be carried out obliviously to protect the variable node state as well as the bit flip information.

From a performance point of view, all steps except the last one are extremely fast in MPCs, given that additions are only local operations and do not need any communication among the MPC peers, and the reductions in step one can be performed in parallel. The oblivious comparisons necessary to decide for each bit (if it has to be flipped or not) are the costly operations and limit the throughput in MPC implementation. However, modern highly optimized MPC systems such as MP-SPDZ (https://github.com/data61/MP-SPDZ (accessed on 20 January 2023)) are able to achieve good performance even for this task, as the results in Table 1 show. These results indicate that MPC-based real-time decoding for QKD is possible.

Clearly, to achieve the best performance, optimized codes must be studied and designed in tandem with MPC protocols [36]. In addition, BP-based alternatives to sum-product decoding should be studied to see how fast MPC versions of belief propagation methods can be pushed. Additionally, for CV-QKD, approximation approaches combined with multi-edge-type codes [22] seem promising for fast MPC implementation. Nevertheless, our experiment already showed the first results and paves the way for practical rates.

Finally, optimization-based decoding would also be possible as an alternative to message-passing algorithms (i.e., by leveraging linear programming (LP)). In LP decoding [37,38], the maximum likelihood decoding problem is formulated as a linear program. Thus, it is possible to decode a symbol by solving an associated LP with conventional approaches (e.g., with a simplex algorithm where MPC versions also exist [39]). However, for the QKD use case with block sizes *k* in the range from $10^{4}$ to $10^{6}$ bits and high error rates, the formulation would lead to a relatively large simplex tableau. Very low rates can be expected for this solution approach, given the measured performance for MPC-based LP solving reported in [40].
**Algorithm 1:** MPC version of bit flip decoding with Gallager’s Algorithm B
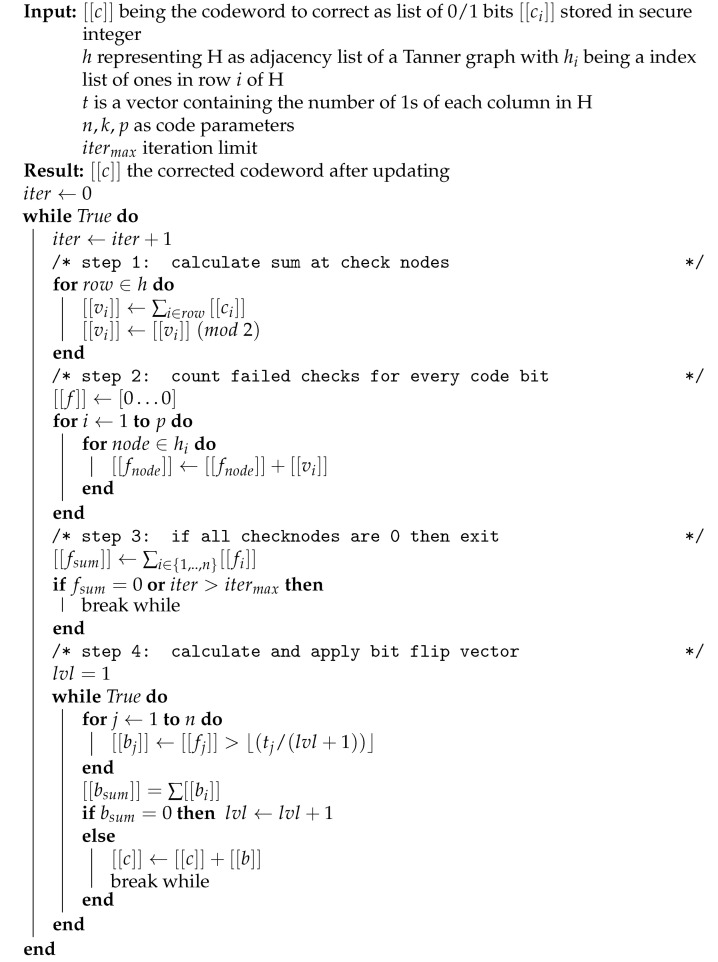


## 4. Offload Privacy Amplification

Privacy amplification (PA) is another important step in the post-processing stack (cf. Section 2.1). This also requires a public channel for communication and is typically based on application of a randomly selected hash of a universal hash family, thus achieving information-theoretical secure randomness extraction. PA is used to extract the mutual information between Alice and Bob such that the adversary Eve is left without any information, except for a negligible error that can be made arbitrarily small.

Although the underlying matrix-vector multiplication seems rather efficient, because of finite key effects and its influence on the secure key rate, a large block length has to be used [29]. Therefore, this step is also computationally very demanding [6], and solutions to entirely offloading this task from the device, or at least from the trusted area within a device, would be desirable.

In a PA protocol, Alice randomly selects a hash from a family of universal hashes with the right compression rate—based on Eve’s potential knowledge on the key—and communicates the selected function publicly to Bob. Both peers then apply the same function on the local reconciled key and arrive at the final shared key. The main property of a universal hash family is that they guarantee a low number of collisions, even if the input is chosen by an adversary.

Because the block length in QKD is large, the complexity of the universal hashes is also relevant. One family of strongly universal hashes is given by multiplication of the raw key with a random matrix which would need a lot of randomness. Nevertheless, to reduce the randomness needed, a Toeplitz matrix can also be used for PA, which requires only n+m random bits compared with the n·m for a random matrix. The use of the Toeplitz matrix also reduces the computational effort for PA because the diagonal structure enables the use of a number theoretical transforms for faster processing of the vector matrix product.

Assume that $k_{A}^{\prime }=k_{B}^{\prime }=k^{\prime }$ is the reconciled key at Alice and Bob with a length *n* and k represents the final keys of a length *m*. Then, PA works as follows:Alice randomly generates a uniform string of a length n+m−1 defining the Toeplitz matrix T and sends it to Bob.Alice computes $k=k^{\prime }\;T$ as the final key.Bob receives T from Alice and also computes his key as $k=k^{\prime }\;T$.

Thus, both parties perform the same vector-matrix multiplication to shrink the identical keys from *n* to *m* bits, where the ratio n/m for CV-QKD is computed as demonstrated by Leverrier et al. [29] and for DV-QKD as shown by Scarani et al. [25].

If we want to offload PA, we would have to offload the core vector-matrix multiplication, which reduces the *n* raw key bits to *m* final bits. The ratio is already known at the beginning of the PA step, but the Toeplitz matrix has to be generated for each block and exchanged clearly, which makes offloading a problem. If T could be kept secret, it would be possible to use a permutation-based approach for PA offloading to a single server, but for a public matrix, this is not possible. However, encrypting T is not an option, as it can be immediately seen that the key balance becomes negative if the encryption key for the matrix is longer than the raw key processed. Thus, hiding the input and output keys while still offloading the computation is not feasible in a single-sever model.

However, if multiple servers are available, a very efficient non-interactive multiparty protocol is possible (i.e., without requiring the servers to communicate). The protocol is shown in Figure 3. The peer shares the raw key in *n* parts with a linear secret sharing scheme—working over $F_{2}$ or a larger prime field $F_{p}$—and sends them to the servers (one share per server). The servers compute the $\left[\left[k^{\prime }\right]\right]=\left[\left[k\right]\right]T$, where [[·]] denotes the sharing of a value. Because of the linearity of the secret sharing scheme, the necessary multiplications with public constants and additions can be carried out on the shares without interaction between the servers. The results are then sent back to Alice, who reconstructs the final key. For the case of prime fields, Alice additionally reduces the result vector mod2.

The security of the protocol against a passive adversary is governed by the security of the underlying secret sharing scheme. Because the parties do not interact with each other but only communicate with the peer, they cannot learn any information about the final key as long as ITS linear secret sharing is used (e.g., additive or Shamir secret sharing [32]). The computational effort of the solution is the same for every server, which would also be the same as for local computation.

Unfortunately, the REM-PA protocol does not provide efficient verifiability aside from recomputation or spot checking, and therefore efficient protection against active attackers cannot be easily achieved during the PA phase. However, if the confirmation round is shifted to after PA, then it will detect errors in the keys and prevent from erroneous keys by aborting the protocol. Thus, it can also detect malicious behavior by the external servers but not directly attribute the errors to them. If the confirmation is shifted to after PA, then it is important to encrypt the tag sent because otherwise, the leaked information cannot be removed anymore, as is normally carried out by PA. Additionally, a secure channel is assumed to distribute the shares to the severs, which may prevent from certain use cases. However, contrary to the MPC-based LDPC decoding, no interaction between servers is required.

## 5. Use Cases

To answer why offloading computationally intensive tasks is interesting at all, we present the expected benefits in general and discuss advantages for certain networking scenarios.

The overall goal which can be achieved is savings in energy or cost at the device side, beneficially impacting the cost-effectiveness of the end user’s equipment. Therefore, if the devices are simpler and require less computational power, then the cost savings could be substantial (e.g., in cases where the user buys the equipment). Compared with data center environments, the devices are also less energy-efficient for running computation-intensive tasks. If this part can be offloaded to a more efficient data center, then the overall operational cost can also be lowered. Therefore, dedicated cloud solutions which further pool information reconciliation for a larger amount could further help to reduce energy consumption. By regularly updating the external hardware resources, the system can benefit from Moore’s law and the continuous drop in cost of computation resources. They can even be shifted flexibly between different locations and data centers to optimize energy usage and cost if more offerings are available. In general, it would even be possible to leverage public cloud services for REM-IR, which requires no trust assumption at all for the environment. Because of these arguments, we think the ability to offload and relocate computationally intensive tasks also leads to higher energy efficiency for computation resources.

Additionally, it could also lead to more flexibility on the QKD level (i.e., QKD as a service). The virtualization of the computationally expensive post-processing tasks could be convenient in the future. Not all optical network units (ONUs) have access to QKD functionality, but the same hardware may be used for coherent passive optical networks (PONs) and CV-QKD [41,42].

Therefore, we may provide QKD to them by just allocating additional processing resources while also switching their software-defined transceiver into QKD mode. Furthermore, networking equipment is installed for longer times and not updated often. This is even more true for high-cost security-certified equipment, because upgrading security-certified equipment is a cumbersome and costly process which typically requires recertification. Being able to update certain non-critical components without needing to exchange or recertify the core QKD device hardware can greatly simplify the upgrade process.

To show how the advantages relate to concrete use cases, we will quickly mention three examples.

*Access networks.* In the case of access networks, we find constraint resources (computing energy), and the network units must be low-cost because they are the driving cost factor, especially since only very low key rates are typically required (AES key refreshing). If computational resources are pooled in such a scenario, costs can be substantially reduced not just in case of a reduced user subscription ratio but also due to time sharing of centralized CPU resources. Furthermore, because the optical part is low energy, the dominant cost and energy factor is when a CPU is partially idle, which should be avoided.

*Satellite communication.* Satellites have a particularly long lifetime (20–30 years) and have to be remotely operated and maintained. They also have limited access to energy resources and reducing energy consumption is of paramount interest. This is especially true if low-cost (mobile) earth stations should be supported or even inter-satellite links. Offloading post-processing can make satellite transceivers possible and increase the connectivity for individual satellites.

*Integrated COTS Hardware.* Finally, aside from the evident advantages of outsourcing protocols such as REM-IR to data centers, the concept can also be interesting when applied within the device. QKD devices are complex systems [43] and comprise many different components, which makes security auditing and certification very hard. To achieve strong security guarantees, only trustworthy hardware and software can be used to process key material in plaintext [44]. Furthermore, to prevent from side channel attacks and backdoors, it would be desirable to reduce the number of trusted components and the complexity of the secure environment in a device as well as possible. Therefore, if the components processing sensitive key materials can be reduced, this results in a smaller attack surface, simplifies security analysis and helps in the certification process. The MPC-based protocols presented can be used for this purpose (i.e., to reduce the trusted environment on the device architecture level with all its benefits). Within a device, it is also feasible to realize the secure channels required in the MPC model. Thus, it would allow for the integration of COTS hardware in QKD systems only processing keys in encrypted form.

## 6. Conclusions

In this work, we introduced the idea of offloading the information reconciliation and privacy amplification steps of QKD post-processing. These are the two computationally intensive tasks in processing raw key measurements to secure a shared key between two QKD peers. We showed that outsourcing information reconciliation is possible and straightforward for DV-QKD, even in a single-server model and against an active adversary. However, for CV-QKD, which leverages reverse reconciliation to overcome the 3 dB transmission bound, the same is not true. We also gave an intuition that it is not possible in general to achieve positive key rates with a single server and analyze potential performance in a multi-server setting. We also looked into privacy amplification, where we propose a protocol for multiple servers. Finally, we laid out the potential benefits and discussed use cases where this approach is relevant.

Proving the impossibility of single-server PA offloading as well as *weak offloading* is left for future work. Additionally, MPC-optimized versions of sum-product decoders are currently under investigation and will be presented in a follow-up work.

## 7. Patents

The basic scheme *IC-REM* from this work was first patented in Austria (AT519476B1) and later in Europe (EP3607446B1) and the US as well (US11128445B2). However, only in this work did we provide the security analysis and additional methods as well as the limitations for the technology.

## Figures and Tables

**Figure 1 entropy-25-00226-f001:**
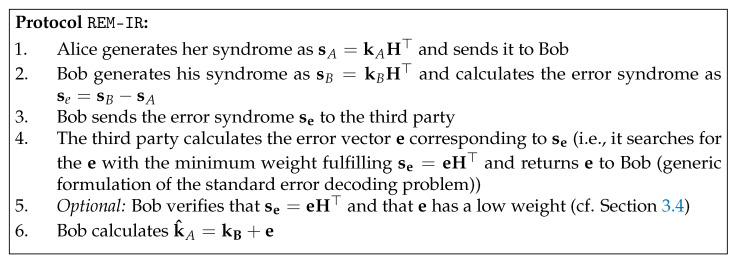
REM-IR protocol.

**Figure 2 entropy-25-00226-f002:**
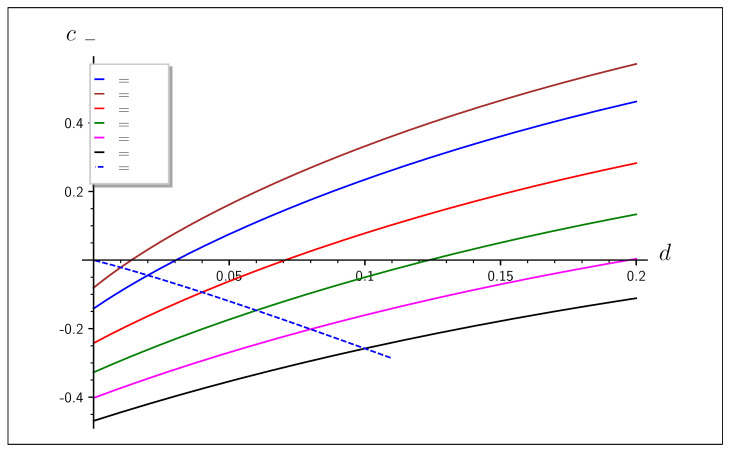
Secret key length balance (secret channel capacity) for reconciliation with encrypted $s_{B}$, enabling error correction offloading to untrusted parties. Values are shown for different *e*. To the left of the dashed line is the interesting region d<e, which has a negative key balance and is therefore unfeasible.

**Figure 3 entropy-25-00226-f003:**
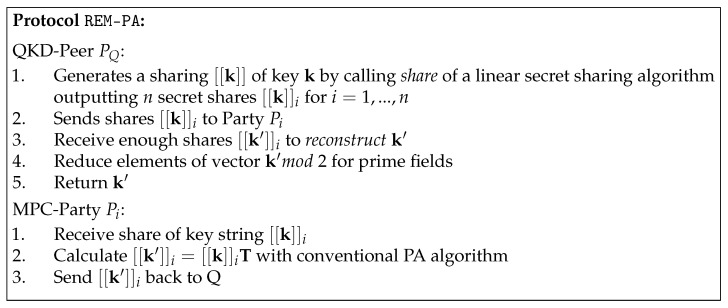
REM-PA protocol.

**Table 1 entropy-25-00226-t001:** Performance comparison of MPC-based oblivious bit flip vector calculation as described in Algorithm 1 (step 4) for LDPC decoding. The measurements were performed in MP-SPDZ with *sharmir.sh* for three nodes with different block sizes and a bitwidth of a secure integer. Circuit depth is then the multiplicative depth of the MPC circuit as processed by MP-SPDZ VMs, and time is the time to compute the circuit. Data represents the amount of information exchanged over the network during rounds of communication required by the VMs as reported by MP-SPDZ. Bitrate@10iter is the estimated throughput which can be achieved based on the measurements. The values show that the kilobit per second regime is feasible even without optimizations and block-level parallelization.

Block Size	Bitwidth	Circuit Depth	Time	Data	Rounds	Bitrate@10iter
			(s)	(MB)		(bps)
1000	4	9	0.06	3.0	65	1571
1000	8	11	0.09	3.8	80	1116
10,000	4	9	0.11	4.4	85	8932
10,000	8	11	0.14	4.7	95	7054
100,000	4	9	1.2	44	805	8354
100,000	8	11	1.4	47	846	7117

## Data Availability

Not applicable.

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
