# Peer review of "On the Security of Offloading Post-Processing for Quantum Key Distribution"

_entropy, 2023, doi:10.3390/e25020226_

Round 1
Reviewer 1 Report
Efficient information reconciliation protocols are computationally intensive. The author proposes a scheme to offload the computational burden to a third party.
I have several comments about the work. I would like to bring up the most important comment first.
1.
In Theorem 1 the author claims that the Protocol REM-IR does not lose any additional information.
However, this cannot be true. The syndrome S_A is already communicated to Bob over a public channel therefore Eve has a copy of it. Additionally, exposing S_e gives bob the syndorm S_B. (The authors note this after the theorem) Since K_A and K_B are not identical (this is why, we are doing information reconciliation), S_A and S_B cannot be identical. So, S_B would expose additional information to Eve. This additional leaked information has to be quantified and later used to perform appropriate Privacy amplification. Therefore, this theorem’s proof is wrong.
In the same proof In line 238, H(K_A)=H(K_B)=1 cannot be true. It should be H(K_A)=H(K_B)=n
2.
The most important concern I have about the work is that the protocol REM-IR is offloading an NP-complete problem to the third party. In the line 4, of Protocl REM_IR the problem of finding e such that s_e=e H^T is the Boolean satisfiability problem that was the very first problem shown to be NP-complete. So having a third party or an oracle that can preform these NP complete computation may have some theoretical interest, but it is practically intractable. Therefore, the protocol is not useful in practice.
The reason why LDPC algorithms work in information reconciliation is that Bob who is performing the LDPC decoding has access to his K_B. This key is guaranteed to have only a small amount of error in comparison to K_A that was estimated during previous QKD steps. Given that the matrix H is a low-density matrix, and Bob has access to K_B, allow Bob to perform the Belief propagation decoding algorithm which is still an approximation algorithm (because the original problem is NP-complete) but works nice in practice. For some cases, the Gallager’s bit flipping algorithm also works. But in the proposed case the third party do not have access to K_B. The Gallager’s Bitflip algorithm has access to the full block. Where the full block contains the erroneous message, here K_B and the syndrome, here S_A. Without this, it is hard to believe that a minimal solution e for s_e=e H^T can be found tractably by flipping bits on block size in the range of thousands (Table 1).
The author may refer to works by Elkouss et al. (doi: 10.1109/ITWKSPS.2010.5503195.) or the ‘Blind Reconciliation’ work (arXiv:1205.5729). for efficient reconciliation techniques for QKD using LDPC.
I would propose that instead of focusing too much on the impossibility of CV-QKD case the author solve the application to DV-QKD more rigorously.
More specifically they need to answer and clearly demonstrate the following.
In a single-party case (where the computation is offloaded to a single third-party) how would the decoding algorithm (for example Gallagar’s bitflip approach) work without access to K_B? Please give more detailed breakdown of the decoding steps.
The Table 1 gives numbers for the Multi-party case. But the case is not properly formulated in the associated discussion in section 3.5. The author needs to be more specific about the algorithm and give pseudocode of their approach/protocols.
3.
The main concern about the offloading of PA is that if the servers collude then they will be able to recover the error corrected key. So, Alice and Bob’s shared key will not remain secret any more. If they do not collude, then this becomes an additional assumption in this proposed Protocol REM-PA.
Moreover, the authors do not give any details of the secret-sharing algorithm used. Is it quantum safe? An eavesdropper with quantum computing capability may break this protocol.
If the author tries to give a formal proof of correctness of this protocol these weaknesses will become self evident.
4.
The description in line 192-197 is not clear. In line 195 the authors talk about compressing data, which is wrong. Syndrome calculation is not the same as compression.
5.
There are some typos (for example, `undermine’ in line 51) and miswording that I am not listing here.
In the light of the above comments, I would not accept this article. The author must rigorously handle the comments 1,2, and 3 and update their manuscript accordingly before the scientific merit of this article can be reconsidered.
Author Response
Please, find our response in the document attached. Br

Reviewer 2 Report
In the manuscript, the authors consider the possibility of offloading the classical post-processing procedures of quantum key distribution (QKD) to a third-party with powerful classical computer.
The authors first study the feasibility of outsourcing and verifying information reconciliation in both direct reconciliation (usually for DV-QKD) and reverse reconciliation (usually for CV-QKD) in an informational theoretically secure way using a single server. Then, the authors consider to combine the post-processing with the assistance of classical multi-partite computation (MPC) protocols. Finally, the authors also discuss about the feasibility of outsourcing and verifying privacy amplification. Their claims are:
1. The direct reconciliation assisted with a single remote server (REM-IR Fig. 1) is still informationally secure, while the reverse reconciliation is not.
2. The results of information reconciliation can be easily verified, and hence the server can be untrusted.
3. Assume we have a secure MPC protocol, we can then perform information reconciliation and privacy amplification efficiently and securely.
4. The results privacy amplification cannot be verified.
Overall, I think this work is interesting and useful for the practical QKD implementation. In many QKD applications, the ability of offloading the post-processing will help to alleviate the hardware requirements for the QKD communication parties. I have two major concerns:
1. I am not convinced of the proof in Theorem 1, which is now unclear. The authors provide the following arguments:
(i) The required amount of information reconciliation is |S_A|= H(K_A|K_B) = n H_b(e);
(ii) The new key information Eve learned from S_A is bounded by the chain rule of relative entropy: H(K|E) - H(K|E, S_A) |S_A| = n H_b(e);
Based on the two arguments above, the authors claim that (below is a variant of the formula between line 245 and 246):
H(K|E, S_A, S_e) – H(K|E) = H(K|E, S_A) – H(K|E).
I cannot see how this holds from the two arguments above. In general, since the variables K -> S_A -> S_e form a Markov chain, we have:
I(K:S_e|S_A) = H(K|S_A) – H(K|S_A, S_e) = 0.
However, when an extra Eve come into play, I(K:S_E|S_A, E)=0 is not obvious, unless we have Pr(S_e|S_A, E) = Pr(S_e|S_A, E, K).
2. In the protocols which outsource the post-processing with the assistance of MPC protocols, the overall security is mainly limited by the security of MPC. Usually, the classical MPC schemes are not informational theoretically secure like QKD: their security relies on the assumptions on the computational complexity. In this case, I cannot see the meaning of using QKD protocols: why not just using the existing public-key schemes (e.g., RSA, or lattice-based algorithms), which owns the same level of security as the joint QKD-MPC schemes?
If the authors can address my concerns above, I can suggest its publication.

Author Response

(The authors gave the same response as above.)

Reviewer 3 Report
The authors, in the paper entitled “On the Security of Offloading Post-Processing for Quantum Key Distribution”, propose a complementary approach, by presenting novel methods for offloading (or outsourcing) the most expensive parts of the QKD post-processing stack. In particular, they present and analyze protocols for outsourcing information reconciliation to external and untrusted environments, therefore facilitating new application scenarios, e.g., usage in low-power access networks.
My opinion is that the introduction of a third part is not necessary and not introduce any particular novelty. The third party describide in the paper can be identified with the eavesdropper (Eve). The paper in the present form can not be published.
Author Response

(The authors gave the same response as above.)

Round 2
Reviewer 1 Report
Line 196, 197 still uses compressed data, instead of syndrome computation.
Note that, data compression means that the compressed data can be decompressed to reconstruct the original input, with little or no error. However, in an information reconciliation (IR) or error correction code, the syndrome does not contain the whole information of the original data. So, they are fundamentally different.
The paper claims to provide multiple results such as, offloading IR related workload to a third party, impossibility for CV-QKD case, and untrusted node based multiparty IR, etc.
I would like to focus on the first and main claim—offloading IR to third party.
To simply put, the proposed protocol does not work.
There is an exponentially large number of e that would satisfy s_e=eH^T
This is because of the dimension of the H matrix. Where the syndrome size S_e is very small, and e is the same size of the keys (K_A or K_B).
Again, to repeat my previous comment number 2. In an usual implementation the Belief propagation algorithm eventually converges because the decoder has access to K_B which is a very closely matched verion of K_A with some bitflips. The decoding Tanner graph utilizes both the syndrome S_A, and K_B to perform its calculations. Moreover, in a multi round protocol additional shortened and punctured bits are used that helps the algorithm to converge.
In, the authors proposed version, the third-party basically has to guess e, check that the e sastisfies s_e=eH^T, and still, it might not be close to the actual error-vector. The number of possible guesses are exponential in number and the algorithm will never work.
I would propose the authors, to instead of spending more effort on the other claims of the paper, first they consolidate the primary claim.
A straightforward way to understand the criticality of my objections above is to try to implement the protocol that the authors propose. In fact, an implementation is a very basic sanity check for any protocol, especially in the absence of a rigorous poof of convergence. If the authors, try to implement the protocol, they will readily find out that the third party cannot successfully perform a LDPC decoding algorithm without access to K_B. and if K_B is given to the third party then the raw key is leaked, and the protocol fails.
I would urge the authors to try to implement their proposed protocol to see why it is wrong.
Now that we see that the primary claim of the paper is incorrect, I will not comment on the other claims presented.
Author Response
Dear Reviewer,
please find our comments attached.
Best regards.

Reviewer 2 Report
I check the authors reply. I have no further comments and think the manuscript suitable to be published.
Author Response
Thank you very much for your valuable comments and support.
Best regards.